# Social Understanding and Friendships in Children with Attention Deficit/Hyperactivity Disorder or Dyslexia

**DOI:** 10.3390/bs15020216

**Published:** 2025-02-15

**Authors:** Sofia Kouvava, Katerina Antonopoulou, Constantinos M. Kokkinos, Asimina M. Ralli

**Affiliations:** 1Department of Economics and Sustainable Development, Harokopio University of Athens, 17676 Athens, Greece; kantonop@hua.gr; 2Department of Primary Education, School of Education Sciences, Democritus University of Thrace, 68131 Alexandroupolis, Greece; kkokkino@eled.duth.gr; 3Department of Psychology, National and Kapodistrian University of Athens, 15772 Athens, Greece; asralli@psych.uoa.gr

**Keywords:** theory of mind, empathy, friendship quality, ADHD, dyslexia

## Abstract

Social understanding relies upon children’s experience of the world and their communicative interaction with others. Opportunities to engage in cooperative social interaction, such as friendships, can foster the development of social understanding. Children with attention deficit/hyperactivity disorder (ADHD) and dyslexia often have friendships of poorer quality. The present study examined relationships and differences in social understanding and friendship quality in children with ADHD or dyslexia, in comparison to neurotypically developing children (NTD). Participants were 192 primary-school students (*M_age_* = 9.77, *SD* = 1.21) from Attica, Greece. Social understanding was measured with second-order and advanced theory of mind (ToM) tasks, and the Bryant’s Index of Empathy for Children and Adolescents, while friendships were examined with the Friendship Quality Questionnaire. Children with ADHD scored significantly lower across all positive characteristics of friendship quality, empathy and advanced ToM than those with dyslexia, who in turn scored significantly lower than NTD children. Moreover, children with ADHD performed significantly worse in the second-order ToM tasks compared to children with NTD or dyslexia, while NTD children scored significantly lower in the friendship conflict betrayal subscale compared to both ADHD and dyslexia groups. Multiple regression analyses controlling for learning disability status, gender, and age showed that ToM and perceived empathy predict friendship quality characteristics in all groups of children. Our findings contribute to a better understanding of children’s friendship relationships and give insight to parents and professionals involved in children’s education, provision, and mental health care.

## 1. Introduction

Social understanding is a broad concept with social and emotional dimensions, which among others encompasses theory of mind (ToM) and empathy. ToM is the ability to attribute mental states (i.e., beliefs, emotions, desires, etc.) to oneself and to others, and to shape one’s behavior accordingly ([65]). Longitudinal and cross-sectional studies demonstrate that success in ToM tasks at a young age is related to an individual’s later socio-emotional competence ([4]; [8]; [17]), to school performance and adjustment ([46]), and to popularity and interpersonal relationships ([14]; [22]). More specifically, research evidence indicates that a well-developed ToM among close friends facilitates the development of intimacy, cooperation, help, and mutual assistance ([21]; [39]; [84]). Respectively, it has been found that preschool children who had at least one friend showed higher performance on ToM tasks, as compared to those peers who did not have any friends, regardless of their language competence and social acceptance ([66]). Finally, it has been reported that conflict in the safe context of a very close friendship relationship can encourage both friends to practice skills important for their ToM and to consider different points of view ([55]).

Empathy is a social competence ([15]) that plays an essential role in social interactions, in socially acceptable behaviors, and in one’s well-being and adjustment ([31]). Empathy helps to understand another person’s point of view and to experience emotional reactions related to the other person’s emotional state ([18]). Research suggests that empathy contributes to the development and quality of interpersonal relationships throughout life ([69]). Specifically, children who have developed empathy can better control their emotional reactions and exhibit fewer aggressive or antisocial behaviors, are altruistic, more optimistic, and demonstrate positive attitudes towards their peers who choose to be their friends ([53]). Hence, children with greater empathy form easier very close friendships, characterized by intimacy, tenderness, and support ([11]; [80]), while conflicts between friends are rarer and resolved by discussion or mutual compromises ([15]).

Friendships are very significant interpersonal relationships, observed across cultures and times, gender and age, and linked with human nature’s need for affection and support ([1]; [13]). It has been postulated that the existence of friends in children’s lives can predict their psycho-emotional adjustment, school readiness and academic performance, as well as their quality of life ([82]). The presence of friendships in one’s life is equally important to the identity of one’s friends and the qualitative characteristics of their friendship ([33]; [57]). Several studies suggest that high-quality best (i.e., very close) friendships, with more positive and less negative qualitative features, significantly influence both friends’ self-esteem, socio-emotional adjustment, and well-being ([35]; [51]). Conversely, low-quality best friendships may lead one or both friends to feelings of loneliness, anxiety, and depression ([44]), and to difficulties in psycho-social adaptation ([71]), expressed by aggressive, antisocial or even delinquent behaviors ([72]).

Children with neurodevelopmental disorders, such as attention deficit/hyperactivity disorder (ADHD) and dyslexia, often encounter challenges not only in their academic performance and social–emotional development but also in establishing and maintaining social connections within the school environment ([49]; [76]). Given the crucial role of friendships in children’s psychosocial development and overall well-being, this study aims to explore potential associations between friendship quality and social understanding—specifically, ToM and empathy—among primary school children with and without ADHD or dyslexia.

## 2. Literature Review

Previous research has shown that in preschool and middle school children and adolescents with neurotypical development (NTD) both ToM and empathy are related to their social behavior, social position, and interpersonal relationships ([37]; [38]). Thus, positive associations have been reported among ToM, empathy, peer acceptance, prosocial behavior, and conflict resolution between friends ([20]; [54]; [73]). From middle childhood, success with empathy and advanced ToM tasks has been positively associated with social adjustment, sociability, popularity, and friendships characterized by cooperation, support, and mutual help ([8]; [38]; [39]). On the contrary, low performance in empathy and ToM tasks has been linked to moral disengagement, social skills deficits, poor friendship qualitative characteristics, and aggressive or even bullying behaviors in children and adolescents ([28]; [41]; [70]). However, certain findings positively link ToM with aggression, suggesting that children with NTD who have a well-developed ToM can easily adopt aggressive or bullying behaviors in their interpersonal relationships, involving mainly social exclusion, rumor-spreading, gossiping, and manipulation of friends ([9]; [28]).

### 2.1. Theory of Mind, Empathy, and Friendships of Children with ADHD or Dyslexia

Children with ADHD face great difficulties in ToM, mainly in understanding the mental states of both themselves and others ([23]; [68]). Moreover, several studies report severe difficulties ([30]; [87]) or a complete lack of empathy ([12]; [50]) in children and adolescents with ADHD. It has also been suggested that children with ADHD, mainly due to the difficulties they encounter in their language skills, cannot easily perceive the thoughts, intentions, feelings, and beliefs of their peers, so conflict resolution is quite limited ([23]). The aforementioned difficulties have been linked with problems related to interpersonal relationships and social acceptance ([79]), and thus, ADHD has often been characterized as a “social disorder” ([27]). Therefore, it could be suggested that children with ADHD problems in understanding others’ perspectives, thoughts, intentions, emotions, and beliefs, and in having low tolerance levels, result in behaviors that may lead them to rejection by their peers and to few friendships ([6]; [26]; [63]; [67]; [74]; [83]), in which conflict resolution is harder to be achieved, and thus, their friendships dissolve easily ([23]). However, other research evidence suggests that children with ADHD have increased empathy, especially for positive emotions ([6]; [26]). There are also studies underpinning the need to further investigate the relationship between social understanding (ToM and empathy) and ADHD children’s social skills and friendships ([74]).

In addition, children with ADHD encounter social difficulties ([78]), due to either their externalizing problems, such as disruptive, defiant or aggressive behaviors, or to their internalizing problems, such as anxiety, mood disorders or depression ([60]). These behaviors lead children with ADHD to frequently be neglected or rejected by their peers in various social contexts ([77]) and thus, to have few friends ([75]), less stable friendships ([52]; [89]), often characterized by lower qualitative features ([2]; [42]; [58]; [67]). Thus, children with ADHD of both genders often report that their friendships are distinguished by several negative features, such as conflict, rivalry, and aggression ([58]).

Findings regarding children with dyslexia and their performance in ToM tasks are scarce and contradictory. Some studies indicate that children with dyslexia do not differ significantly in their performance in ToM tasks as compared to their counterparts with NTD ([10]), while others consider that children with dyslexia, and especially those with severe reading difficulties, show significantly low performance in ToM tasks ([25]; [40]). In the advanced ToM tasks, though, where the understanding of metaphors, humor, irony, lies, etc., are examined, children with dyslexia were found to have severe difficulties ([19]). Moreover, the very limited research on empathy in children with dyslexia, positively associates phonological awareness and reading ability with empathy ([25]) and agrees that these children have greater difficulties compared to their peers with NTD ([3]; [5]). These difficulties in empathy result in hindering children with dyslexia from acknowledging the feelings of their peers or attaining a realistic view of reality, leading them to problems in their interpersonal relationships ([19]; [25]).

Children with dyslexia state that they have very few best friends and great difficulties in forming and maintaining close friendships with their peers at school ([48]), where because of their learning difficulties, they feel embarrassed, ashamed, insecure, and inferior ([16]; [47]). Furthermore, children with dyslexia also reveal that their friendships have poor qualitative characteristics, lacking support and intimacy ([16]). Quite often they tend to befriend peers with similar difficulties, as in these relationships they feel more accepted and freer to discuss their problems ([86]). However, research focusing on how ToM and/or empathy may affect the friendships of children with dyslexia is, to our knowledge, non-existent. Finally, there are no published studies comparing social understanding in children with ADHD, dyslexia, and NTD, nor exploring the relationships between aspects of social understanding and friendship quality in these three groups of children.

### 2.2. Purpose of the Study

Even though friendships are important bonds across the lifespan and instrumental for individuals’ socio-emotional adjustment, mental health, and well-being, the studies that have examined possible associations among friendships, ToM, and empathy in children with NTD are very limited ([55]). In addition, research evidence linking ToM and empathy with the friendship quality of children with ADHD or dyslexia is scarce ([67]). Thus, the present study will try to enrich and advance scientific knowledge regarding the relationship among ToM, empathy, and the perceptions of friendship quality in children with ADHD, dyslexia, and NTD, by answering the following research questions:Are there any differences in the performance of children with ADHD or dyslexia in empathy, second-order or advanced ToM tasks compared to their peers with NTD?To what extent ToM and empathy are associated with the qualitative characteristics of the best friendships of children with ADHD, dyslexia or NTD?Can social understanding and specifically ToM and empathy predict the positive and negative qualitative aspects of the best friendships of children with ADHD, dyslexia or NTD?

## 3. Materials and Methods

### 3.1. Participants

One hundred and ninety-two children with either ADHD, dyslexia or NTD (64 in each group) participated in the present study. All participants, who were between 8 and 12 years of age (*M_age_* = 9.77 years, *SD* = 1.21), attended randomly selected inclusive mainstream primary schools in Attica, Greece, and had their parents’ consent to take part in the study. No significant differences were observed among the three groups regarding children’s gender and age, as shown in Table 1. As all the participating children with ADHD and dyslexia had an official diagnosis (confirming either ADHD or dyslexia, with no comorbidity), by public educational and counselling centers, supervised by the Greek Ministry of Education, Religion, and Sports, no diagnostic procedures were performed in this study. In all ADHD diagnoses among the children, the specific subtype of the disorder was not specified; therefore, this variable was not included in the analysis of the present study. Children with NTD were selected based on their teachers’ quarterly reports on their academic achievement.

### 3.2. Measures

*Demographics:* Participants answered a short demographic questionnaire with information regarding their gender, date of birth, and school year.

*ToM—Second-order false belief tasks:* To test second-order reasoning, probing children’s ability to handle the beliefs of one person about the beliefs of another, stories such as the ice cream van were used ([64]). Three stories presented in an A4 folder were read to all participants to avoid misunderstandings, due to possible reading difficulties. Each story had 4 pictures depicting its main points. Participants could view the pictures and the script while the researcher read the story. The names of the main characters in all stories were translated or altered to common Greek names, so that the participants could relate with them. One of the stories, for example, is about John and Mary who are in the park, as well as an ice cream van. Mary wants to buy an ice cream, but she does not have any money with her. The man in the ice cream van announces that he is going to be in the park for the afternoon, so Mary goes back home to get some money. After Mary leaves and John is still at the park, the man in the ice cream van tells John that as there are not many people in the park, he will take the van to the church for the rest of the day. However, when at home, Mary sees the ice cream van passing by and finds out that it will be at the church for the rest of the day. As John does not know that Mary knows that the van will be at the church, he believes that Mary believes that the ice cream van is still at the park. Therefore, we think that John will think that Mary thinks that the ice cream van is still at the church. This example of second-order ToM or second-order belief attribution, because we have to access two mental states (John’s mental state of Mary’s mental state) to answer the question. Certain types of questions followed each story: a reality question (e.g., ‘Where did Mary go to buy an ice cream?’) and a memory question (e.g., ‘Where was the ice cream van in the morning?’), to verify that the participants understood the story. An experimental question (e.g., ‘Where does John think Mary went?’) for the identification of the second-order belief was scored 0 or 1 based on its correctness. If the participant could not answer it, a forced question) followed, including two options (e.g., ‘John believes that Mary went to the church or to the park?’). Finally, there was a documentation question to assess the understanding of the second order of belief, as it asked to justify the answer (e.g., ‘Why does John think Mary is there?’). The documentation question was scored 0, 1, or 2, depending on whether there was an incorrect documentation (0 points) or a correct one, using natural (e.g., ‘To buy an ice-cream’) (1 point) or mental (e.g., ‘Because John didn’t know that Mary had talked to the man in the ice cream van and knew that he went to the church’) situations (2 points). Thus, the maximum score for each story was 3.

*Advanced ToM tasks:* The Strange Stories ([32]) were used to examine participants’ advanced metalizing ability. To overcome a possible ceiling effect in second-order false belief tasks which are often fully acquired by the age of 8 or 9 years in children with NTD ([64]), and as the participants’ ages were between 8 and 12 years, twelve advanced ToM tasks requiring the interpretation of non-literal statements in the context of social narratives we administered ([34]). The Strange stories were presented as short vignettes and participants were asked why a character says something that is not literally true. Correct answers require attribution of certain mental states, such as pretense, joke, lie, white lie, misunderstanding, irony, forgetting, double bluff, contradicting emotions, persuasion, and figure of speech ([85]). Each vignette has a title related to its topic in the upper left side of the brochure, together with a small, black-and-white representative image and two questions: a comprehension question (e.g., “Is it true what s/he said?”) and a documentation question (e.g., “Why did s/he say this?”), in the context of which the justification of the answers was ascertained. If a comprehension question was answered incorrectly, the participants were encouraged to listen to that particular vignette again and to look at their script. There was no time limit for answering the comprehension and documentation questions. Correct answers to comprehension questions were scored with 1. Answers to documentation questions were scored with either 0 if they were incorrect, or with 1 (correct—physical documentation) and with 2 (correct—mental state documentation) ([24]). This test has satisfactory internal consistency (α = 0.65) and reliability ([34]). To avoid fatigue and distraction of the participants, and as the partial administration of the mentalistic vignettes does not affect the final result ([34]), children were presented with only one (of the two) vignette for each figure of speech.

*Bryant’s Index of Empathy for Children and Adolescents:* This widely used questionnaire ([7]), validated ([56]) and adapted in Greek ([81]), examines children’s and adolescents’ empathy. For primary school children, the tool consists of 22 items, with ‘Yes’ or ‘No’ as possible answers. Positive answers are scored with 1 and negative answers with 0 (e.g., ‘I really like to watch people open presents, even when I don’t get a present myself’). There is also a reverse scoring for some sentences (2, 3, 9, 10, 15, 16, 17, 18, 20, 21, 22), where ‘No’ is scored with 1 and ‘Yes’ with 0 (e.g., ‘It is hard for me to see why someone else gets upset’). The final score can range from 0 to 22, with 22 indicating the greatest empathy. In the present study, Cronbach’s alpha for the Bryant’s Index of Empathy was 0.64.

*Friendship Quality Questionnaire:* The Friendship Quality Questionnaire (FQQ) ([62]) is a world-accepted tool, designed to examine children’s perceptions about the quality of their best friendships. It has been adapted to Greek ([43]), along with many other languages. In its long form, FQQ has 40 items allocated in the following 6 subscales: (a) validation and caring (e.g., makes me feel good about my ideas), (b) companionship and recreation (e.g., always play together at recess), (c) help and guidance (e.g., share things with each other); (d) conflict and betrayal (e.g., fight a lot), (e) conflict resolution (e.g., make up easily when we have a fight); and finally, (f) intimate exchange (e.g., always tell each other our problems). Validation and caring, companionship and recreation, help and guidance, conflict resolution, and intimate exchange are considered to be the positive qualitative characteristics of the best friendship, while conflict and betrayal is a negative qualitative feature. At the beginning of the administration of the questionnaire, participants were asked to identify and to write on its initial page the name of their best same-sex friend from their classroom and then to answer all the questions having this friend in mind. Answers were given on a 5-point Likert-type scale (not at all true = 0, a little true = 1, somewhat true = 2, pretty true = 3, really true = 4), and the score for each subscale was the average of the ratings for the relevant items. Only item 21 is reverse-scored. In the present study, Cronbach’s alpha was 0.75 for companionship and recreation, 0.90 for help and guidance, 0.84 for conflict and betrayal, 0.73 for conflict resolution, 0.90 for validation and caring, and finally, 0.86 for intimate exchange, and for whole FQQ was 0.96.

### 3.3. Procedure

Initially, the Greek Institute for Educational Policy granted permission for the present research, and then the head teachers and the board of teachers of inclusive mainstream primary schools in Attica, Greece, were approached and informed about the purpose of the study. In the schools that agreed to participate, parental written consent was asked and when provided, all the questionnaires and tasks were administered individually to children. All questionnaires and tasks were administered individually to each child. To ensure comprehension, every questionnaire and task was thoroughly explained, and all items were read aloud, particularly to support children with reading difficulties. Additionally, if a child exhibited signs of fatigue or restlessness, the session was briefly paused to allow for a break or, if necessary, rescheduled to continue on another day. Teachers provided a code number for each participant to ensure their anonymity. The whole procedure, which took place in a secluded room (e.g., library) on the school premises, lasted about one hour.

## 4. Results

One-way ANOVA was performed to examine group differences in friendship quality, second-order false belief understanding, advanced ToM, and empathy. Pearson correlations were used to test for possible correlations among the different variables. Finally, a single hierarchical regression analysis was computed for the friendship variables, in order to examine major determining factors including social understanding (ToM and empathy), group, age and gender. All statistical analyses were performed with IBM SPSS version 24 statistical software.

### 4.1. Research Question 1: Are There Any Differences in the Performance of Children with ADHD or Dyslexia in Empathy, Second Order or Advanced ToM Tasks Compared to Their Peers with NTD?

Table 2 presents differences in the friendship quality characteristics (both positive and referring to the negative characteristics of conflict and betrayal in friendships) and in the social understanding variables (2nd order false belief understanding, advanced ToM and empathy) among ADHD, dyslexia and NTD groups. Post-hoc comparisons have shown that ADHD children tend to experience friendships of significantly poorer quality (less validation and caring, less help and guidance, less companionship and recreation, less intimate exchange, and less conflict resolution) than the other two groups (*p* < 0.05), and performed significantly worse on all social understanding tasks than their NTD peers (*p* < 0.01). Similarly, children with dyslexia had friendships of lower quality than their NTD peers (*p* < 0.05) and their performance on social understanding was worse than that of their NTD counterparts (*p* < 0.01). However, the dyslexia group did not differ from the ADHD group in 2nd order false belief understanding. Additionally, NTD children were more likely to experience less conflict and betrayal in their friendships than were children with dyslexia and ADHD children (*p* < 0.05).

### 4.2. Research Question 2: To What Extent ToM and Empathy Are Associated with the Qualitative Characteristics of the Best Friendships of Children with ADHD, Dyslexia or NTD?

In order to examine possible associations among friendship quality aspects and social understanding outcomes, all positive attributes of friendship quality (validation/caring, conflict resolution, help/guidance, companionship/recreation, intimate exchange) were averaged to compose the ‘friendship quality-positive attributes’ variable, whereas the negative aspect of friendship quality, that of ‘conflict-betrayal’ remained a separate variable named ‘friendship quality-conflict/betrayal’. Similarly, ‘2nd order false belief understanding’ and ‘advanced ToM’ made a total sum ToM score with empathy remaining a separate score, both reflecting social understanding. Positive correlations were found among the friendship quality-positive attributes, ToM and empathy for each group separately and for the total number of participating children. Significant negative correlations emerged among ToM and conflict/betrayal in friendship quality for the ADHD, the NTD group and for the total number of participants but not for the dyslexia group. Finally, empathy was negatively associated with conflict/betrayal for the ADHD group and the total number of participants (Table 3).

### 4.3. Research Question 3: Can Social Understanding and Specifically ToM and Empathy Predict the Positive and Negative Qualitative Aspects of the Best Friendships of Children with ADHD, Dyslexia or NTD?

To examine whether variance in children’s friendship quality would be partially explained by the existence or not of learning disability (group status), age, gender and social understanding (ToM and Empathy), a hierarchical regression analysis was computed separately for each of the two friendship quality variables (Positive attributes, Conflict-betrayal). Age and gender were entered in Step 1, group (ADHD, Dyslexia and NTD) in Step 2, and the two social understanding variables (ToM and Empathy) in Step 3 (Table 4).

For the positive attributes of friendship quality, the existence of a learning disability predicted 22% of the variance (Step 2), showing that the absence of a learning disability can predict better friendship quality, but when social understanding variables (both ToM and empathy) were entered into the model the prediction was raised to 54% of the variance with better social understanding predicting higher positive quality in friendships for all the groups of children. Age, gender and group did not significantly predict the positive qualities of friendships in the final model. As regards the negative aspect of friendship quality that of conflict and betrayal in Step 2 group was found to negatively predict 11% of the variance, meaning that NTD children experienced less conflict and betrayal in their close friendships than the ADHD/dyslexia groups. In Step 3, ToM but not empathy added another 14% to the prediction, explaining in total 25% of the variance.

## 5. Discussion

The present study sought to examine possible differences in the performance of primary school children with ADHD, dyslexia or NTD in social understanding, and particularly in ToM and empathy, and how these differences might be associated with the qualitative characteristics of their best friendships. It also attempted to examine whether ToM and empathy could predict the positive and negative qualitative characteristics of the best friendships of children with and without ADHD or dyslexia.

Initially, children with ADHD reported that their best friendships were characterized by poorer positive (i.e., validation and caring, companionship and recreation, help and guidance, conflict resolution, and intimate exchange) and greater negative (i.e., conflict and betrayal) qualitative features. When compared to the other two groups, i.e., children with dyslexia and NTD, participants with ADHD had the lowest scores in all the positive and the highest scores in the negative friendship characteristics. The results confirmed previous findings suggesting that children with ADHD tend to have great difficulties in forming and maintaining high-quality friendships ([2]; [67]). Specifically, research evidence reveals that children with ADHD claim that their best friendships are distinguished by more negative qualitative features, and thus they are not particularly satisfied with them ([42]; [58]). The present study furthers our knowledge by providing evidence of the differences in the perceptions between children with ADHD and dyslexia regarding the quality of their friendships, as previous research on this field is lacking.

Significant differences were also observed in the qualitative features of their best friendships between children with dyslexia and NTD, with the former perceiving that their friendships have more negative and less positive characteristics. These findings are consistent with previous research, both international and Greek ([16]; [48]), revealing that children with dyslexia also face difficulties in their friendships and consider them to be of low quality. A possible explanation for the children with dyslexia perceiving their best friendships to be of poor quality may be due to their feelings of insecurity and inferiority, and their self-awareness of their difficulties at school ([16]; [47]), as they are always guarded and conceal their problems from their peers with NTD. Consequently, their friendships are distinguished by the lack of support and intimacy ([16]; [47]). This argument is quite plausible, as children with dyslexia prefer to befriend their peers with similar difficulties with whom they feel free to discuss their preoccupations ([86]).

The present results verify previous findings stating that children with either ADHD or dyslexia face difficulties in their friendship relationships with their peers ([52]; [60]; [75]; [77]; [89]). As published research focusing on comparisons between these two groups’ friendships is to our knowledge non-existent, only suggestions could be made regarding the differences found in our study. However, the differences that emerged regarding the friendship quality of children with ADHD or dyslexia can be readily explained. While both ADHD and dyslexia are neurodevelopmental disorders, they have different manifestations and severity of symptoms. Thus, these children’s difficulties in forming and maintaining qualitative best friendships may have different origins. More specifically, children with ADHD can exhibit disruptive, defiant or aggressive behaviors ([78]), or can be inattentive to their friends’ needs ([60]), leading to peer rejection ([77]) and to less stable friendships ([52]; [89]), characterized by lower qualitative features, such as validation and caring, companionship and recreation, help and guidance, conflict resolution, and intimate exchange ([2]; [42]; [58]; [67]). On the contrary, children with dyslexia who also report having few friendships with more negative and less positive qualitative characteristics do not invest in their friendships or pursue the best friendships with high quality, as they often feel insecure or inadequate as individuals because of their poor academic performance ([16]; [47]).

When compared to both ADHD and dyslexia groups of children, the best friendships of children with NTD were characterized by significantly fewer negative features, i.e., conflict and betrayal. This is in agreement with previous findings ([28]; [37]) regarding the friendships of children with NTD, who usually seek to have and enjoy supportive and caring friendships and try to overcome conflicts and quarrels in order to maintain them. On the other hand, children with ADHD or dyslexia due to their externalizing (i.e., disruptive, defiant or aggressive) or internalizing (i.e., anxious, embarrassed or depressive) behaviors experience more conflicts and betrayal in their friendships ([42]; [48]; [58]). As to our knowledge, there are no previous studies comparing the negative qualitative characteristics in the best friendships of children with ADHD and dyslexia, and the present work provides evidence that both groups of children have equally great difficulties in dealing with conflict with and betrayal from their friends.

Regarding the differences in social understanding, children with ADHD and dyslexia performed worse in all tasks compared to their peers with NTD. More specifically, in all the variables examined, children with ADHD scored significantly lower than the children with dyslexia, who in turn, scored significantly lower than the children with NTD. However, the only exception was the second-order ToM tasks, where no differences were observed between children with ADHD and dyslexia. These findings are in accordance with previous work stating that children with ADHD have severe difficulties both in ToM ([23]; [68]) and empathy ([30]; [87]), which result in problems in their social interactions ([79]). Moreover, it could be suggested that the lowest performance in all social understanding tasks from the part of children with ADHD could be attributed to the core characteristics of the disorder, mainly inattention or impulsivity, which may affect children’s reactions to the thoughts, beliefs, emotions, and desires of others. Regarding the children with dyslexia, our results verify those studies which postulate that children with dyslexia show significantly lower performance in (second-order false belief and advanced) ToM ([19]; [25]; [40]) and empathy ([3]; [5]) tasks, when compared to children with NTD. As no research has focused so far on the comparison between children with ADHD and dyslexia in ToM and empathy, only cautious speculations could be made for the similarity in their performance (in the second-order false belief tasks) that emerged in this study. It could be postulated, therefore, that as second-order false belief tasks are quite demanding, requiring a child’s ability to predict someone’s belief about what someone else thinks about a situation, they are not fully acquired by the age of 8 or 9 years ([64]). As our sample consisted of children between the ages of 8 and 12 years, those children with ADHD and dyslexia might have a slower pace in the development of their ToM. However, further exploration is required, as the difficulties exhibited in ToM by children with ADHD and dyslexia might be attributed to the problems they experience in their social environment (e.g., rejection by peers, etc.) or in emotion regulation/control.

The present study also revealed positive associations between social understanding (ToM and empathy) and the positive qualitative features of children’s friendships for all groups. This finding is to be expected, considering that when children with or without NTD acknowledge the others’ points of view and emotions when they are together having fun or when they negotiate to overcome a conflict, their friendships would be long-lasting and of high quality. Research supports this finding, as it has recognized that, predominantly in children with NTD, social understanding can facilitate the development of intimacy, cooperation, help, and mutual assistance between friends ([21]; [39]; [84]), who are able to control their emotional reactions and to challenge their opinions in order to understand the other’s perspective ([55]), exhibiting more altruistic behaviors and positive attitudes towards their peers and friends ([53]). The present study contributes to the relevant research by attesting that ToM and empathy play an important role in the positive friendship qualitative features of children with ADHD and dyslexia.

Negative correlations were found between ToM and the negative qualitative features of children’s friendships, i.e., conflict and betrayal, for the ADHD and NTD groups and for all the participating children, but not for the dyslexia group. Breaking down this finding, it could be claimed that for the children with NTD, a well-developed ToM, as expected in the participants of this study according to their age, would facilitate the resolution of the conflicts with those friends who deem to be close to them and appreciate their friendships ([55]). However, regarding the children with ADHD, who have great difficulties in their ToM, this result could imply that the less they take into account the intentions, desires, and feelings of their friends, the more arguments they have in their friendships ([63]; [67]; [83]) which dissolve easier, as conflict resolution is harder to be achieved ([70]). Finally, some hypotheses can be made about the absence of associations between ToM and conflict and betrayal in the friendships of children with dyslexia, as this field has not been adequately investigated. Considering the difficulties that children with dyslexia face in all (second-order false belief and advanced) ToM tasks ([19]; [25]; [40]), they might be lacking or feeling inadequate when they confront conflicts in their friendships. Thus, they might prefer to either skip a fight with their friends or end a friendship that is not fulfilling.

Empathy was found to be significantly negatively associated with the negative qualitative features of the friendships of children with ADHD. The children in this group are prone to have difficulties in empathy ([30]; [87]), leading to problems in their interpersonal relationships ([42]; [79]), among which are their friendships ([42]; [58]; [67]), and thus, ADHD has often been characterized as a “social disorder” ([27]). Specifically, low performance in empathy has been linked to social skills deficits, low friendship qualitative characteristics, and aggressive or even bullying behaviors in children and adolescents with or without NTD ([35]; [44]; [51]).

Moreover, the present findings establish important relationships between friendships’ qualitative characteristics and social understanding in all the participating children. The absence of disability predicted higher positive qualitative features in children’s friendships, regardless of their age and gender. When both ToM and empathy entered the equation, the prediction of the positive quality of friendships for all the groups of children was even higher. Thus, the more the children acknowledge and understand the beliefs, intentions, desires, and feelings of their friends, the better quality they can experience in their friendship relationships. This finding is in accordance with several longitudinal and cross-sectional studies in children with NTD indicating that high performance in ToM and empathy tasks enhance the quality of their interpersonal relationships ([14]; [22]; [39]; [84]). However, this finding also suggests that when children have a well-developed social understanding, the existence of a neurodevelopmental disorder might not gravely affect the positive qualities of their best friendships.

Finally, the absence of disability could moderately negatively predict the negative aspect of friendship quality, while ToM but not empathy can increase the percentage of this prediction. Mainly, this implies that children with NTD experience less conflict and betrayal in their close friendships than children with ADHD or dyslexia, and this is reinforced by a well-developed ToM (second-order false belief and advanced), but not by empathy. This finding agrees with research evidence regarding ToM, which postulates that when confronted with a conflict, children with NTD have to challenge their own opinions and beliefs, in order to understand and accept their friends’ points of view ([55]). As the analysis of the data in the present study failed to find evidence of the role of empathy in the prediction of the negative qualitative characteristics of children’s friendships, further exploration in this field is required. However, a preliminary suggestion could be that at this age, children might not have fully developed their empathy skills, and thus, they are not always able to experience emotional reactions related to their friends’ emotional state ([18]).

## 6. Limitations, Future Directions, and Conclusions

The present study has certain limitations. First, as the focus of our research was children’s friends at school, friendships were examined exclusively at a classroom level, ignoring other relevant relationships they may have. Future research could explore children’s friendships in various social environments (e.g., neighborhood, extra-curricular activities). Second, the study was solely based on self-report instruments and thus, the results may be vulnerable to informant bias. To overcome this obstacle and to have more objective results, in future research, information could be gathered from various sources (e.g., peers, parents, teachers, observation). Third, participant selection for the ADHD and dyslexia groups was based exclusively on the diagnoses submitted to their respective schools without additional assessments to verify the presence or severity of each disorder. The provided diagnoses identified either ADHD (without specifying the subtype or the symptoms of the disorder) or dyslexia, with no reported comorbidities. Thus, no associations could be made between ADHD symptoms and friendship qualitative features. Future research should explore how the presence of co-occurring disorders or variations of symptoms across ADHD subtypes may influence the observed outcomes. Moreover, it would also be interesting to examine possible relationships between the symptoms of children’s disorders and their social skills. Finally, a longitudinal experimental design could provide insight into changes in social understanding due to children’s development and maturation, possibly leading to different results regarding the friendship quality characteristics in the three groups of children.

Despite its limitations, this study contributes to the advancement of scientific knowledge by providing significant evidence linking social understanding (both ToM and empathy) to the qualitative features of children’s best friendships. Although there is scientific evidence regarding ToM, empathy, and friendship quality in children with NTD, relevant research particularly for children with ADHD or dyslexia is very limited and sometimes controversial. Taking into consideration the significance of friendships in children’s social adjustment, and every future interpersonal relationship, the findings of the present study are very important for both children with and without NTD. Although it could be argued that our research results are somewhat predictable, as children with ADHD or dyslexia often face not only learning but also social challenges at school, they are nonetheless important because they link children’s difficulties with their peers to certain factors, such as ToM and empathy. As similar studies are very limited, the contribution of our findings is quite important, extending our knowledge in this field. They may also have practical implications, as they could inform and sensitize the school community and the policymakers, leading to the acknowledgment of the need to apply educational programs and evidence-based strategies in mainstream schools or training interventions in clinical settings that could enhance the social understanding (ToM and empathy skills) and to reinforce long-lasting, high-quality friendship bonds, especially in children with ADHD, dyslexia or any other neurodevelopmental disorder. Integrating ToM and empathy-building interventions into existing school curricula can be carried out through structured activities that align with standard educational and social–emotional learning (SEL) frameworks. For example, such interventions could include: (a) storytelling and narratives ([29]; [61]), focusing on the main characters’ emotions and perspectives; (b) discussion of social scenarios with real-life challenges and hypothetical social situations ([36]) (e.g., a friend feeling left out, a misunderstanding, how to resolve conflicts, etc.); (c) structured role-playing activities ([88]) (e.g., enacting a social dilemma and reflecting on how different responses impact our relationships); or even (d) integrating ToM and empathy into existing SEL programs (e.g., identification and understanding of emotions in oneself and others, using emotion charts, videos, or AI-driven tools like facial expression recognition apps) aiming at fostering social understanding and improving friendship quality. In addition, peer-mediated interventions at school have been found to promote perspective-taking, collaboration, learning, and communication, as socially skilled peers can model appropriate social behaviors ([45]; [59]). Finally, an important factor that ensures the success of intervention programs at school and at home is the continuous training of teachers and parents on the strategies that encourage students’ perspective-taking and empathy-building.

## Figures and Tables

**Table 1 behavsci-15-00216-t001:** Demographics by group.

		NTD (n = 64)		Dyslexia (n = 64)		ADHD (n = 64)		Differences	
		*f*	%	*f*	%	*f*	%	*x* ^2^	*p*
Gender	Boy	31	48.4	33	51.6	32	50	0.13	0.94
	Girl	33	51.6	31	48.4	32	50		
Grade level	3rd	16	25	16	25	16	25	0.21	0.99
	4th	17	26.6	15	23.4	16	25		
	5th	16	25	17	26.6	16	25		
	6th	15	23.4	16	25	16	25		
		*M*	*SD*	*M*	*SD*	*M*	*SD*	*F* _(2,104)_	*p*
Age (in years)		9.67	1.15	9.82	1.15	9.81	1.16	0.29	0.75

Note: NTD = children with neurotypical development, Dyslexia = children with dyslexia, ADHD = children with attention deficit/hyperactivity disorder.

**Table 2 behavsci-15-00216-t002:** Group comparisons of friendship quality and social understanding.

	NTD		Dyslexia		ADHD		ANOVA		
	*M*	*SD*	*M*	*SD*	*M*	*SD*	*F* _(2,191)_	η^2^	Post Hoc Comparisons
Friendship Quality									
Validation-Caring	4.50	0.49	3.37	0.79	3.05	0.89	67.35 *	0.42	ADHD < Dyslexia < NTD
Conflict-Betrayal	1.82	0.43	2.37	0.67	2.51	0.71	22.48 *	0.19	NTD < ADHD, Dyslexia
Conflict Resolution	4.41	0.68	3.52	0.97	3.05	0.94	40.69 *	0.30	ADHD < Dyslexia < NTD
Help-Guidance	4.18	0.72	2.94	1.02	2.47	0.99	59.11 *	0.39	ADHD < Dyslexia < NTD
Companionship-Recreation	4.55	0.47	3.36	0.89	3.01	0.90	68.28 *	0.42	ADHD < Dyslexia < NTD
Intimate Exchange	4.16	0.81	3.12	1.02	2.62	1.04	42.96 *	0.31	ADHD < Dyslexia < NTD
2nd order false belief	7.78	2.61	3.23	3.03	2.19	2.69	73.16 **	0.44	ADHD < Dyslexia, NTD
Advanced ToM	25.61	4.52	18.39	4.26	16.06	3.33	95.87 **	0.50	ADHD < Dyslexia < NTD
Empathy	14.75	2.77	9.64	5.17	7.53	4.41	49.01 **	0.34	ADHD < Dyslexia < NTD

Note. ANOVA = analysis of variance; NTD = children with neurotypical development, Dyslexia = children with dyslexia, ADHD = children with attention deficit/hyperactivity disorder; ToM = theory of mind. * *p* < 0.05, ** *p* < 0.01—Bonferroni correction of critical *p* values when performing post hoc multiple comparisons.

**Table 3 behavsci-15-00216-t003:** Correlations among friendship quality-positive attributes, friendship quality-conflict/betrayal, theory of mind and empathy in children with ADHD, dyslexia and neurotypical development.

	ToM (2nd Order False Belief and Advanced ToM)				Empathy			
	Dyslexia (n = 64)	ADHD (n = 64)	NTD (n = 64)	Total Sample (N = 192)	Dyslexia (n = 64)	ADHD (n = 64)	NTD (n = 64)	Total Sample (N = 192)
Friendship Quality—Positive Attributes	0.40 **	0.35 **	0.54 **	0.69 **	0.40 **	0.38 **	0.42 **	0.62 **
Friendship Quality—Conflict-Betrayal	−0.23	−0.31 *	−0.25 *	−0.48 **	−0.06	−0.26 **	−0.15	−0.36 **

* *p* < 0.05, ** *p* < 0.01. Note: NTD = children with neurotypical development, Dyslexia = children with dyslexia, ADHD = children with attention deficit/hyperactivity disorder; ToM = theory of mind.

**Table 4 behavsci-15-00216-t004:** Summary of hierarchical regression analysis for the prediction of friendship quality-positive attributes and conflict-betrayal in friendships.

Variable	β	*t*	*F*	*R* ^2^	Δ*R*^2^	Δ*F*
Friendship Quality—Positive Attributes						
Step 1			1.16	0.12	0.12	1.16
Age	0.04	0.57				
Gender	0.10	1.42				
Step 2			17.54	0.22	0.21	49.72 *
Age	0.06	0.99				
Gender	0.09	1.42				
Group	0.45	7.05 *				
Step 3			44.04	0.54	0.32	65.66 *
Age	−0.04	−0.76				
Gender	0.05	1.01				
Group	0.08	1.32				
ToM (2nd order false belief and Advanced ToM)	0.48	6.46 *				
Empathy	0.28	4.23 *				
Friendship Quality—Conflict-Betrayal						
Step 1			0.04	0.01	0.01	0.04
Age	−0.01	−0.14				
Gender	0.02	0.25				
Step 2			7.87	0.11	0.11	23.52 *
Age	−0.03	−0.38				
Gender	0.03	0.39				
Group	−0.33	−4.85 *				
Step 3			12.08	0.25	0.13	16.46 *
Age	0.06	0.85				
Gender	0.04	0.59				
Group	−0.08	−1.09				
ToM (2nd order false belief and Advanced ToM)	−0.38	−4.03 *				
Empathy	−0.09	−1.13				

* *p* < 0.01.

## Data Availability

The original contributions presented in this study are included in the article. Further inquiries can be directed to the corresponding author.

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
