# Peer review of "Social Understanding and Friendships in Children with Attention Deficit/Hyperactivity Disorder or Dyslexia"

_behavsci, 2025, doi:10.3390/bs15020216_

Round 1

Reviewer 1 Report

Comments and Suggestions for Authors

This study focused on the friendship quality in children with ADHD or dyslexia, and its relations with empathy and mind theory. The topic is interesting. However, there are still some issues in the current version of the manuscript as below:

--What are the possible explanations to the inconsistent findings regarding the friendship quality among ADHD or dyslexia? The authors should provide the possibilities about the conflicting findings. Based on this analysis, the authors could design the present study, and to examine whether it is exactly the reason for the inconsistency. Not just do a study without clear analysis, that will make the inconsistency more puzzled.

--What is the rationale to examine the friendship quality among children with ADHD and dyslexia? Besides the first group comparisons, datasets of children with ADHD and dyslexia were analyzed in combination of that of typically developing children, without revealing any special pattern in the special children. The authors should emphasize the importance of examine the issue.

--What are the subtypes of ADHD in the present study? Please provide detailed information about this issue. Do most of children with ADHD belong to ADHD-I or ADHD-c OR ADHD-H? Maybe children with ADHD-H have more problems in friendship than the other two subtypes.

--Because poor friendship quality among children with ADHD and dyslexia result from their neurodevelopmental disorder to a great extent, what is the relationship between these social skills and their symptoms? The authors should supplement relevant analyses, which might bring some enlightments for examining the friendship quality in these special-needed students.

Reviewer 2 Report

Comments and Suggestions for Authors

The study aimed to examine the relationships and differences in social understanding and friendship quality among children with ADHD, dyslexia, and neurotypically developing (NTD) children. In light of the study’s necessity and importance, its foundation is meaningful. However, several areas require attention and improvement:

1. Separation of introduction and literature review: The introduction and literature review should be clearly distinguished. The introduction must provide a focused and concise rationale for conducting the study, presenting its significance without being conflated with the literature review. Meanwhile, the literature review should thoroughly discuss prior research, highlight gaps, and justify the current study’s design. Currently, these sections are mixed, and the review of previous studies is insufficient. To strengthen the manuscript, detailed information explaining the rationale for the research and its design should be relocated to the literature review.

2. Participants and survey administration: The study included children with ADHD, dyslexia, and NTD. Administering surveys to children with ADHD and dyslexia presents significant practical challenges due to the nature of these conditions. To ensure the reliability of the study results, the manuscript must provide a detailed description of the measures taken to address these challenges. For instance, it should explain how tasks and questionnaires were tailored to accommodate attention deficits, impulsivity, or reading difficulties, thereby ensuring valid and credible data collection.

3. Educational and psychological implications: The study demonstrates that children with ADHD and dyslexia face friendship challenges due to a lack of social understanding. It also suggests that enhancing ToM and empathy skills can improve the quality of their friendships, offering significant implications for educational and psychological interventions. While the findings are valuable, they may seem somewhat predictable. To ensure the study contributes meaningfully to the field, it is crucial to propose specific, actionable interventions. For example, the manuscript could outline targeted strategies, such as ToM training programs or empathy-building activities, and discuss how these could be implemented in educational or clinical settings.

4. APA reference style: Based on my understanding, the manuscript’s references, both in-text and in the reference list, must adhere to APA reference style. Corrections are necessary to meet academic standards and ensure consistency. 

Comments on the Quality of English Language

The study aimed to examine the relationships and differences in social understanding and friendship quality among children with ADHD, dyslexia, and neurotypically developing (NTD) children. In light of the study’s necessity and importance, its foundation is meaningful. However, several areas require attention and improvement:

1. Separation of introduction and literature review: The introduction and literature review should be clearly distinguished. The introduction must provide a focused and concise rationale for conducting the study, presenting its significance without being conflated with the literature review. Meanwhile, the literature review should thoroughly discuss prior research, highlight gaps, and justify the current study’s design. Currently, these sections are mixed, and the review of previous studies is insufficient. To strengthen the manuscript, detailed information explaining the rationale for the research and its design should be relocated to the literature review.

2. Participants and survey administration: The study included children with ADHD, dyslexia, and NTD. Administering surveys to children with ADHD and dyslexia presents significant practical challenges due to the nature of these conditions. To ensure the reliability of the study results, the manuscript must provide a detailed description of the measures taken to address these challenges. For instance, it should explain how tasks and questionnaires were tailored to accommodate attention deficits, impulsivity, or reading difficulties, thereby ensuring valid and credible data collection.

3. Educational and psychological implications: The study demonstrates that children with ADHD and dyslexia face friendship challenges due to a lack of social understanding. It also suggests that enhancing ToM and empathy skills can improve the quality of their friendships, offering significant implications for educational and psychological interventions. While the findings are valuable, they may seem somewhat predictable. To ensure the study contributes meaningfully to the field, it is crucial to propose specific, actionable interventions. For example, the manuscript could outline targeted strategies, such as ToM training programs or empathy-building activities, and discuss how these could be implemented in educational or clinical settings.

4. APA reference style: Based on my understanding, the manuscript’s references, both in-text and in the reference list, must adhere to APA reference style. Corrections are necessary to meet academic standards and ensure consistency. 

Round 2

Reviewer 1 Report

Comments and Suggestions for Authors

The authors have made efforts on revising the manuscript. I still look forward to the results about the relationship between the friendship of ADHD children and their symptoms. 

Author Response

Reviewer 1

Comments and Suggestions for Authors

Comment: The authors have made efforts on revising the manuscript. I still look forward to the results about the relationship between the friendship of ADHD children and their symptoms.

Response: We appreciate the reviewer's feedback regarding the revisions we made in our manuscript based on his/her very insightful comments!

Addressing your comment after the second revision of our paper, and as no associations could be made between the friendships of children with ADHD and their symptoms (the official diagnoses of the participating children did not include the subtype of the disorder nor the symptoms they had), we added a sentence in the limitations of the study (page 14, lines 555-557) and in the suggestions for future research (page 15, line 559). The new additions are in green color, while the previous ones are in red.

The provided diagnoses identified either ADHD (without specifying the subtype or the symptoms of the disorder) or dyslexia, with no reported comorbidities. Thus, no associations could be made between ADHD symptoms and friendship qualitative features.”

“Future research should explore how the presence of co-occurring disorders or variations of symptoms across ADHD subtypes may influence the observed outcomes.”

Reviewer 2 Report

Comments and Suggestions for Authors

The reviewers' comments have been largely addressed. However, this study investigates the relationship between friendship quality and social understanding in children with and without ADHD or dyslexia within a school setting, and the research results are somewhat predictable (expected). Furthermore, although similar studies are very limited and conducting experiments in this area is challenging, there remains some uncertainty regarding the extent to which these findings can meaningfully contribute to the relevant field.

Comments on the Quality of English Language

The reviewers' comments have been largely addressed. However, this study investigates the relationship between friendship quality and social understanding in children with and without ADHD or dyslexia within a school setting, and the research results are somewhat predictable (expected). Furthermore, although similar studies are very limited and conducting experiments in this area is challenging, there remains some uncertainty regarding the extent to which these findings can meaningfully contribute to the relevant field.

Author Response

Reviewer 2

Comments and Suggestions for Authors

Comment: The reviewers' comments have been largely addressed. However, this study investigates the relationship between friendship quality and social understanding in children with and without ADHD or dyslexia within a school setting, and the research results are somewhat predictable (expected). Furthermore, although similar studies are very limited and conducting experiments in this area is challenging, there remains some uncertainty regarding the extent to which these findings can meaningfully contribute to the relevant field.

Response: We appreciate the reviewer's feedback regarding the revisions we made in our manuscript based on his/her very insightful comments!

Addressing your comment after the second revision of our paper, we added in the section of Conclusions (page 15, lines 572-577) the following sentences (see text in green color).

“Although it could be argued that our research results are somewhat predictable, as children with ADHD or dyslexia often face not only learning but also social challenges at school, they are nonetheless important because they link children’s difficulties with their peers to certain factors, such as ToM and empathy. As similar studies are very limited, the contribution of our findings is quite important, extending our knowledge on this field.”

Moreover, to stress the meaningfulness of our findings and their contribution to the school community, the following phrases were also added (page 15, lines 578-579) (see text in green color, adjustments from round 1 review are in red):

“They may also have practical implications, as they could inform and sensitize the school community and the policy makers, leading to the acknowledgement of the need to apply educational programs and evidence-based strategies in mainstream schools or training interventions in clinical settings, that could enhance the social understanding (ToM and empathy skills) and to reinforce long-lasting, high-quality friendship bonds, especially in children with ADHD, dyslexia or any other neurodevelopmental disorder.”
